# Cannabis Use in Physicians: A Systematic Review and Meta-Analysis

**DOI:** 10.3390/medicines10050029

**Published:** 2023-04-27

**Authors:** Pierre-Louis Naillon, Valentin Flaudias, Georges Brousse, Catherine Laporte, Julien S. Baker, Valentin Brusseau, Aurélie Comptour, Marek Zak, Jean-Baptiste Bouillon-Minois, Frédéric Dutheil

**Affiliations:** 1Université Clermont Auvergne, CNRS, LaPSCo, CHU Clermont–Ferrand, WittyFit, F-63000 Clermont-Ferrand, France; pierre-louis_naillon13@hotmail.fr (P.-L.N.);; 2Université de Nantes, Laboratoire de Psychologie des Pays de la Loire, LPPL, F-44000 Nantes, France; 3Université Clermont Auvergne, NPsy-Sydo, CHU Clermont–Ferrand, Addiction, F-63000 Clermont-Ferrand, France; 4Université Clermont Auvergne, Clermont Auvergne INP, CNRS, Institut Pascal, CHU Clermont-Ferrand, F-63000 Clermont-Ferrand, France; 5Sport and Physical Education, Hong Kong Baptist University, Kowloon CN-99230, Hong Kong; 6Université Clermont Auvergne, CHU Clermont-Ferrand, Endocrinology Diabetology and Metabolic Diseases, F-63000 Clermont-Ferrand, France; 7INSERM, CIC 1405 CRECHE Unit, CHU Clermont-Ferrand, F-63000 Clermont-Ferrand, France; 8Institute of Health Sciences, The Jan Kochanowski University of Kielce, P-25-002 Kielce, Poland

**Keywords:** addiction, marijuana, healthcare professional, public health, prevention

## Abstract

**Background**: Cannabis use by physicians can be detrimental for them and their patients. We conducted a systematic review and meta-analysis on the prevalence of cannabis use by medical doctors (MDs)/students. **Method**: PubMed, Cochrane, Embase, PsycInfo and ScienceDirect were searched for studies reporting cannabis use in MDs/students. For each frequency of use (lifetime/past year/past month/daily), we stratified a random effect meta-analysis depending on specialties, education level, continents, and periods of time, which were further compared using meta-regressions. **Results**: We included 54 studies with a total of 42,936 MDs/students: 20,267 MDs, 20,063 medical students, and 1976 residents. Overall, 37% had used cannabis at least once over their lifetime, 14% over the past year, 8% over the past month and 1.1 per thousand (‰) had a daily use. Medical students had a greater cannabis use than MDs over their lifetime (38% vs. 35%, *p* < 0.001), the past year (24% vs. 5%, *p* < 0.001), and the past month (10% vs. 2%, *p* < 0.05), without significance for daily use (0.5% vs. 0.05%, NS). Insufficient data precluded comparisons among medical specialties. MDs/students from Asian countries seemed to have the lowest cannabis use: 16% over their lifetime, 10% in the past year, 1% in the past month, and 0.4% daily. Regarding periods of time, cannabis use seems to follow a U-shape, with a high use before 1990, followed by a decrease between 1990 and 2005, and a rebound after 2005. Younger and male MDs/students had the highest cannabis use. **Conclusions**: If more than a third of MDs tried cannabis at least once in their lifetime, this means its daily use is low but not uncommon (1.1‰). Medical students are the biggest cannabis users. Despite being common worldwide, cannabis use is predominant in the West, with a rebound since 2005 making salient those public health interventions during the early stage of medical studies.

## 1. Introduction

Cannabis dependence is one of the most common drug use disorders [1]. Several studies have described cannabis use in the general population [2,3]. Even if there are some studies on drug issues in medical doctors [4,5,6], there are very few studies and heterogenous data on cannabis use by physicians. In the general population, the 2012 European annual report on drug use showed that 25% of the 15–64 year-old Europeans tried cannabis at least once in a lifetime, 6.8% used cannabis during the past year, 3.6% during the past month and 1% had a daily use [7]. However, there is no such prevalence of cannabis use synthesized for medical doctors. Despite a very high percentage of cannabis use among students [8], data are scarce in the medical field and there are a lack of comparisons with medical doctors. Some medical specialties may also be more prone to cannabis use because of working conditions such as stress at work and workload, for example [9,10,11]. Moreover, the use of cannabis is widely heterogeneous across the globe, depending on culture and specificities of regions of the world [12]. The trends towards a decrease in the use of cannabis over time is common in most regions [12]. However, no study has focused on the regional and time effect of the use of cannabis in medical doctors or medical students. Lastly, some sociodemographic features such as age and sex are common influencing factors of cannabis use—men traditionally being more frequent users than women [13].

Thus, we aimed to conduct a systematic review and meta-analysis on the prevalence of cannabis use by physicians, to assess the prevalence and frequency of use, to determine if the cannabis use happens once over a lifetime or more frequently, such as daily cannabis use. The secondary objectives were to report physicians’ cannabis use stratified by specialty, educational level (medical students or medical doctors), continents, periods of time and putative influencing factors such as age and sex.

## 2. Methods

### 2.1. Literature Search

The PubMed, Cochrane Library, PsycInfo, ScienceDirect and Embase databases were searched for entries up until 30 March 2022 with the following keywords: addict* AND (physician* OR doctor*) AND (marijuana OR cannabis). Details for the search strategy within each database are available in Appendix A. To be included, studies needed to describe our primary outcome variable, i.e., the use of cannabis in physicians. We considered any frequency of cannabis use (from once in a lifetime to daily use). The search was not limited to specific years or languages. Reference lists of publications that met our inclusion criteria were manually searched to retrieve further articles. Two authors (Pierre-Louis Naillon and Jean-Baptiste Bouillon) conducted the literature searches, reviewed the abstracts, and, based on the selection criteria, decided the suitability of the articles for inclusion, and extracted the data. When necessary, disagreements were solved with a third author (Frédéric Dutheil) (Figure 1). This systematic review was performed according to the Preferred Reporting Items for Systematic Reviews and Meta-Analysis (PRISMA) guidelines (Appendix A). The study was not registered, as there were delays in procedures during the COVID-19 pandemic [14].

### 2.2. Data Extraction

The primary outcome was cannabis use in physicians (once over a lifetime/past year/past month/daily). Secondary outcomes were education level (student or doctor), medical specialty, country and continent, period of the study, and sociodemographic (age, gender and family status).

### 2.3. Quality of Assessment

We used the Newcastle–Ottawa Scale (NOS) to check the quality of included articles [15]. The maximum score was nine for the cohort and ten for the cross-sectional studies. Additionally, we also used the Strengthening the Reporting of Observational Studies in Epidemiology (STROBE) for cohort and cross-sectional studies (Appendix A) [16].

### 2.4. Statistical Considerations

Statistical analysis was conducted using Stata software (v16, StataCorp., College Station, TX, USA). Extracted data were summarized for each study and reported as mean (standard deviation) and number (%) for continuous and categorical variables, respectively. The prevalence of cannabis use and 95% confidence intervals (95% CI) were estimated using random effects models assuming between- and within-study variability (DerSimonian and Laird approach) [10,17,18]. More specifically, we conducted four meta-analyses on the use of cannabis in physicians (once over a lifetime/past year/past month/daily). Then, for each meta-analysis, we stratified results depending on level of study (student, resident, or doctor), specialties, continents, and periods of time (before 1990, between 1990 and 2005, and after 2005). Statistical heterogeneity between studies was assessed using forest plots, confidence intervals and I^2^. The I^2^ statistic is the most common metric for measuring heterogeneity and is easily interpretable: heterogeneity is considered low for I^2^ < 25%, modest for 25–50%, and high for >50%. We aimed to conduct a sensitivity analysis by excluding studies not evenly distributed around the base of the metafunnel. We also proposed meta-regressions to investigate putative factors influencing the prevalence of cannabis use in physicians, such as level of study, specialties, continents, periods of time, and sociodemographic (age and gender). Results were expressed as regression coefficients and 95 CI. Type I error was fixed at a = 0.05.

## 3. Results

An initial search produced 3958 possible articles. The removal of duplicates and use of the selection criteria reduced the number of articles reporting cannabis use among physicians to 54 articles in the systematic review and 52 articles in the meta-analysis (Figure 1), because two articles focused only on addicted physicians [19,20]. The main characteristics of the studies are presented in Table 1. We describe below the articles included in the meta-analysis.

### 3.1. Quality of Articles

Using the NOS criteria, the studies demonstrated a low risk of bias, except for response bias and insufficient description of statistical tests in the cross-sectional studies and for ascertainment and adequacy of follow up for the longitudinal study (Figure 2). Results were similar using STROBE (Appendix A). Details for each study are available in Table 1.

### 3.2. Study Designs and Objectives

All studies were cross-sectional [8,21,22,23,24,25,26,27,28,29,30,31,32,33,34,35,36,37,38,39,40,41,42,43,44,45,46,47,48,49,50,51,52,53,54,55,56,57,58,59,60,61,62,63,64,65,66,67,68,69], except for two cohorts [70,71]. All the 52 studies included described cannabis-use prevalence among physicians or medical students. The main objective was to assess the prevalence of multiple-substance use in physicians in most studies [8,20,21,22,23,25,26,27,28,29,30,31,33,34,35,36,39,40,41,42,43,44,45,47,48,49,50,52,53,55,56,57,58,59,60,61,62,63,64,65,66,67,69,70,71]. Six studies assessed health status and psychological wellbeing of physicians [19,30,37,40,51,68], four focused on cannabis-use prevalence and belief about cannabis among physicians [24,32,46,54], and one on physicians’ attitudes toward drug testing [38]. 

### 3.3. Recruitment of Physicians

Medical doctors were recruited randomly using the quota method and stratification, using a national or state physician database before mailing or a phone call [27,33,35,37,47,48,58], from lists of diplomas delivered from faculties [29,38,39,45,70], by mailing all physicians from a country/state using an inter-university or national database [31,34], and at a meeting in a medical school [46]. Students and residents were recruited in medical schools—either monocentric [8,21,22,23,25,26,28,40,42,49,50,51,52,53,54,57,59,61,62,63,64,65,66,67,69,70,71] or multicentric [24,30,32,36,41,44,55,60,68]—and at a convention or a festival [43,46].

### 3.4. Populations Studied

Sample size ranged from 46 [25] to 5426 [33,48]. In total, 42,297 physicians or medical students were included in this meta-analysis: 20,267 medical doctors (15 studies) [27,29,31,33,34,35,37,38,39,45,46,47,48,58,70], 20,063 medical students (37 studies) [8,21,22,23,24,25,26,28,30,32,36,40,42,43,44,46,50,51,52,53,54,55,56,57,59,60,61,62,63,64,65,66,67,68,69,70,71], and 1976 residents (two studies) [41,49].

Age was reported in 26 studies. Overall, the mean age was 25.74 years old (95 CI 22.67 to 28.81), ranging from 20 [68] to 53 [37] years old.

Gender was reported in 84% of the studies (*n* = 44). The mean percentage of men was 59% (95 CI 55 to 64%), ranging from 91% [45] to 29% [40]. Cannabis use by gender was described in 13 studies [26,28,30,41,42,44,51,52,56,57,62,68,70] (Appendix A).

Specialty was mostly not reported in 86% of the studies (*n* = 45), followed by anesthesiologists (four studies, *n* = 5075) [33,34,39,58], and general practitioners (three studies, *n* = 1269) [27,31,33].

The location of studies was always reported. Most studies were conducted in North America (twenty-five studies, *n* = 23,903), followed by Europe (twelve studies, *n* = 9963), South America (nine studies, *n* = 4685), South Asia (four studies, *n* = 3095) and Oceania (two studies, *n* = 651).

Other variables were less well described. Family status was reported in 12 studies [25,34,35,36,37,38,39,41,45,48,55,66]. The study level when first use of cannabis occurred was recorded in four studies [8,28,41,44].

### 3.5. Cannabis Use Assessment

Most studies used a self-administered questionnaire (postal or email, after a brief explanation of the study goal) (35 studies) [8,21,22,24,25,26,29,30,31,32,33,34,35,36,37,38,39,40,41,42,43,44,45,46,47,48,49,53,54,55,58,59,63,67,69]. Other studies collected data from an interview or class interview [23,28,51,52,56,57,60,61,62,64,66,68,71], from an interview and by post [70], and from a phone call [27]. The data collection method was unclear in two studies [50,65]. The definition of cannabis use was never described in included articles; however, we can assume a smoking use. 

### 3.6. Frequency of Use and Period of Data Collection

A total of 32 studies (*n* = 29,521 physicians/medical students) reported cannabis use at least once over a lifetime [8,21,23,24,25,26,27,28,29,30,32,35,38,39,40,41,42,43,44,45,46,47,48,49,50,51,52,53,54,55,57,58,60,61,62,64,65,66,67,68,69,70], 26 studies (*n* = 26,500) over the past year [8,21,22,26,28,29,31,32,33,34,35,36,37,41,44,48,50,52,55,56,57,59,63,69,70,71], 20 studies (*n* = 17,341) over the past month [8,24,26,27,28,29,35,36,41,43,44,48,52,55,57,59,60,66,69,70], and 12 studies (*n* = 11,172) daily [8,24,26,28,30,35,41,48,55,59,63,66]. Publication occurred within two years of data collection for 54% of studies [21,22,24,25,27,28,29,32,35,36,37,39,40,42,45,47,48,50,51,52,56,58,62,67,68,69,70], within 2 to 5 years for 31% [26,30,31,34,38,41,43,44,46,49,53,54,55,59,61,66,71], and more than 5 years for 8% [23,33,57,60], and was not reported for 7% of studies [63,64,65]. Studies ranged from 1971 [24,25,32] to 2021 [58], with 15 studies before 1990 (*n* = 9137) [8,21,24,25,28,29,32,42,46,47,50,55,61,62,65], 18 between 1990 and 2005 (*n* = 21,756) [23,27,33,34,35,38,39,41,44,45,48,49,51,64,66,68,70,71], and 19 after 2005 (*n* = 11,404) [22,26,30,31,36,37,40,43,52,53,54,56,57,58,59,60,63,67,69] (Table 1).

### 3.7. Meta-Analysis on the Prevalence of Cannabis Use in Physicians

Taking physicians and medical students together, 37% (95 CI 31 to 43%) tried cannabis at least once in a lifetime, 14% (12 to 17%) smoked cannabis at least once in the past year, 8% (6 to 9%) in the past month, and 0.11% (0.0 to 0.2%) daily. Whatever the frequency of cannabis use (once over a lifetime/past year/past month/daily), medical students seem to have a greater cannabis use than medical doctors: 38% (30 to 45%) vs. 35% (26 to 44%) over a lifetime, 24% (19 to 29%) vs. 5% (3 to 8%) over the past year, 10% (7 to 13%) vs. 2% (0 to 4%) over the past month, and 0.5% (0.1 to 0.9%) vs. 0.05% (0.0 to 0.1%) for daily use (Figure 3). 

Too few studies detailed medical specialties, precluding robust conclusions. Our meta-analysis showed that 41% (38 to 44%) of anesthesiologists and 20% (15 to 26%) of general practitioners tried cannabis at least once over a lifetime, and 3% (2 to 3%) of anesthesiologists and 5% (3 to 6%) of general practitioners over the past year. Studies conducted in North America and Europe seem to have a high percentage of cannabis users among physicians and medical students: 47 and 31% tried cannabis at least once in a lifetime, respectively; 13 and 25%, respectively, in the past year, 10 and 10%, respectively, in the past month, and 0.5 and 0.7%, respectively, daily. Asian countries reported a low percentage of cannabis use: 16% over a lifetime, 10% in the past year, 1% in the past month, and 0.4% daily. South America seemed to report an intermediate prevalence of cannabis use in physicians and medical students. Regarding periods of time, cannabis use seems to follow a U-shape, with a high use before 1990, followed by a decrease between 1990 and 2005, and a rebound after 2005: 22, 9, and 18% for the past-year use, 10, 6, and 8% for the past-month use, and 2, 0.1, and 0.4% for daily use in relation to <1990, 1990–2005, and >2005, respectively. For lifetime consumption, 40, 38, and 31% tried cannabis (<1990, 1990–2005, >2005, respectively). For each stratification, all I^2^ were high (>80%) (Figure 3 and Figure 4).

### 3.8. Meta-Regressions

Medical students had a higher cannabis use than medical doctors over a lifetime (coefficient 0.19, 95 CI 0.09 to 0.29), over the past year (0.18, 0.11 to 0.25), and over the past month (0.07, 0.01 to 0.13). There was no difference whatever between specialties in the frequency of use (once over a lifetime/past year/past month/daily). Over the past year, Europe tended to have a higher prevalence of cannabis use in physicians/medical students than North America (0.12, −0.02 to 0.26). South Asia had or tended to have a lower prevalence of cannabis use over the past month than North America (0.09, 0.01 to 0.17), Europe (0.08, −0.01 to 0.18), and South America (0.07, −0.01 to 0.16). For the past year, cannabis use tended to be the lowest between 1990 and 2005 (coefficient 0.12, 95 CI 0.00 to 0.24 vs. <1990, and 0.08, −0.01 to 0.18 vs. >2005). Male physicians had or tended to have a higher rate of cannabis use over the past year (0.37, −0.05 to 0.79) and over the past month (0.19, 0.04 to 0.33) than women. Younger physicians tended to have a higher rate of cannabis use over the past year (coefficient −0.08, 95 CI −0.18 to 0.01 per 10 years). There were no other significant results or tendencies for all other putative explaining variables, whatever the frequency of use considered (once over a lifetime/past year/past month/daily) (Figure 5) (Appendix A).

## 4. Discussion

The main findings were that the prevalence of cannabis use in medical doctors/students over a lifetime is high, at around 37%. Daily use was rare but not uncommon, with 1.1‰ of medical doctors/students smoking daily. Medical students have the greatest use. Despite the fact that cannabis use is common both in developed and developing countries, there are some cultural differences, with a predominant use in the West. After a decrease in cannabis use after the 1990s, there has been a rebound since 2005. Young and male physicians seem to have higher cannabis use.

### 4.1. Cannabis Use by Physicians: A Public Health Issue

Cannabis is not a benign substance, and inhalation of cannabis smoke is more harmful than tobacco smoke, delivering 50 to 70% more carcinogens [72]. Cannabis also decreases the immune function, promotes cardiac arrythmias and anxiety, and can lead to schizophrenia for genetically predisposed people [72,73,74]. Cannabis can also exacerbate pre-existing psychosis [75]. Medical doctors are exposed to many stressors, from long working hours, sometimes at night, to life-and-death emergencies [76]. Stress may lead to addictive behavior [77,78,79], and consequently the medical profession seems more subjected to drug abuse and psychiatric disorders [80]. Cannabis may be used by physicians to decrease their stress, such as in post-traumatic stress syndrome [81]. Interestingly, consumers using cannabis as a stress-coping strategy are those with the greatest risk of addiction [82,83]. Unfortunately, no studies included in our meta-analysis reported the workload. Despite not studied for cannabis, medical doctors who smoke tobacco promote less cessation advice to patients [84,85,86]. The impact of the use of cannabis by medicals doctors on their practice warrants further studies. Quitting cannabis is quite hard, with less than 10% success at 6 months [87]. The predominant psychotropic component is Δ^9^ tetrahydrocannabinol (THC), and the major non-psycho-active ingredient is cannabidiol (CBD). Both THC and CBD are a partial agonist or antagonist of prototypical cannabinoid receptors CB1 and CB2 [88]. No pharmacotherapy treatments demonstrated efficacy—from nicotine replacement therapy to psychotropic drugs [89]—but there are effective psychosocial interventions [89], such as completing self-determined goals [87,90]. Very interestingly, targeting the microbiome as a therapeutic and diagnostic tool may also be a promising avenue of exploration in the forthcoming years, considering the role of the gut–brain axis in a wide range of substance-use disorders [91].

### 4.2. From Daily to Lifetime Use

Our results showed a logical decrease from lifetime, to year, to month, to daily use, in accordance with the literature [8,24,26,28,35,41,48,55,57,63,66]. In the general population, 25% of the 15–64 year-old Europeans tried cannabis at least once in a lifetime, 6.8% used cannabis over the past year, and 3.6% over the past month [7]. Results from our meta-analysis suggest a higher prevalence of cannabis use by physicians for lifetime and monthly use, which may be linked with work-related stress [76]. Cannabis use has already been studied in other stressful jobs, such as US military veterans who reported a 12% use over the past 6 months [92]—our results for physicians were still higher. Cannabis use may also influence cognitive performance. For example, cannabis multiples by two the risk of a fatal road accident [93]. We note that no study assessed whether cannabis was used at work or during the rest periods of physicians. The moment of use may be relevant when considering the putative side effects of cannabis on medical errors. Cannabis use by physicians is relevant to fitness-for-duty concerns, similar to those for other drugs and alcohol [94,95,96]. We also demonstrated that physicians with a daily use of cannabis were very rare and uncommon (0.1%), compared with the 1% of daily users in the general population [7], which may reflect a low percentage of addicted physicians. Moreover, cannabis addiction may be more linked to the amount of cannabis than to the frequency of use [97]. The amount of cannabis smoked has not been studied in any of our included studies, and could be a salient indicator.

### 4.3. Medical Students as the Heaviest Consumers

We demonstrated a greater use of cannabis in medical students compared to medical doctors, which could be in line with the desire for new experiences in youngers [98]. Medical students may also use cannabis at university to cope with stress or depressive episodes [99]. However, cannabis decreases memory function in students (and cannabis abstinence leads to improved memory), and is associated with the poorest academic performance [99], which could be related to the cannabis-induced hypodopaminergic anhedonia [100]. Early first use of cannabis is also a risk factor for schizophrenia and bipolar disease [101,102]. Following the example of successful alcohol prevention university [103,104], efficient preventive strategies should take place in universities. Nonetheless, all physicians should benefit from targeted preventive strategies. We did not show differences in cannabis use between specialties, but specialties were seldomly reported, which precluded robust conclusions. The most frequently reported specialties were anesthesiologists [33,34,39,58] and general practitioners [27,31,33]. Anesthesiologists are known to be at greater risk of use of psychoactive substances, probably due to overwork and easier access to drugs [105]. If general practitioners are the heaviest tobacco smokers [10], data are lacking for cannabis. To our knowledge, there are no data on co-addiction in medical doctors, i.e., the combination of smoking, alcohol, cannabis or other psychoactive drugs. As alcohol use may predict cannabis use, particularly in the youngest [106], a longitudinal follow-up may be of particular interest.

### 4.4. Cannabis Use Worldwide and through Time

We showed huge disparities among continents in the use of cannabis by physicians, following the trends of cannabis use in the general population [107]. Differences between continents may be explained by a complex interplay between laws, cultural and religion beliefs [108]. In the USA, cannabis use is more frequent in states that legalized cannabis [109], but the legal status has not been reported in our included studies. In Europe, cannabis legislation also differs widely among countries [110]. South America seems to occupy the middle ground of cannabis use in physicians, with huge differences in cannabis policies. Chile and Uruguay have legalized recreational use, while Peru and Bolivia have harsh laws restricting both medical and recreational use [26,111]. Most of our included studies from South America were from Brazil [52,53,56,57,58,60,63], which authorizes cannabis for both personal use and medical use [112]. The continent of Asia continent very repressive policies, explaining the low prevalence of cannabis use [111]. Paradoxically, cannabis was used in Central Asia from 12.000 years ago, and Ayurvedic medicine first used it in India 3.000 years ago [113]. The decline in cannabis started all around the world in the early 20th century, after the second International Opium Convention. The opium and drugs trade were restrained and regulated [113]. Several Western countries then “tolerated” cannabis and reintroduced permissive laws from the 1960s [113]. In Oceania, Australia legalized recreational cannabis use in 2016, and New Zealand is thinking about legalization [50,114]. We included only two studies from Oceania [40,50], but physicians seem to follow the general population of Oceania [107]. A UNO worldwide report showed the same trend in cannabis use by continents as our results, in particular with America and then Europe being the biggest users, far above Asia [107]. Even if there is no study from Africa in our meta-analysis, the same UNO report showed a high cannabis use in Africa, at levels between America and Europe [107]. Despite 1.4 billion people, Africa has less than 1% of all scientific articles on addiction [115]. Regarding periods of time, we demonstrated that cannabis use seems to follow a U-shape. Cannabis use was high before 1990, and so the decrease from the 1990s is coincident with the laws and regulations on cannabis and other drugs [116], followed by a rebound after 2005 that could be a consequence of more permissive laws [117]. The perception of cannabis evolved as a “non-risky” recreational use [117]. In the USA, the states that legalized medical usage of cannabis saw an increase in illicit cannabis use [118]. The consequences of legalization of cannabis are still under debate. As medical use of cannabinoids has become more available, and the need for an evidence-based evaluation of safety and efficacy is necessary [119].

### 4.5. Other Influencing Variables

Male physicians tend to have higher rate of cannabis consumption than women, in our study. In the general population, men use cannabis more frequently and in a higher quantity than women [13]. The effect of cannabis use is different, according to sex. Male users report improved memory, enthusiasm, musicality, and increased appetite, whereas women have a desire to clean and a loss of appetite [13]. We also demonstrated that a younger age is linked with cannabis use, in line with the high consumption of medical students. 

Unfortunately, body mass index, physical activity, marriage, or co-addictions were lacking in the studies included in our meta-analysis. However, cannabis users are less likely to suffer from obesity [120] or to have low levels of physical activity [121,122]. Marriage was associated with a reduction in drug use—including cannabis [123]. Smoking tobacco increases the risk of illicit-drug use such as cannabis [124], and early cannabis use has been strongly associated with other illicit-drug use [125].

### 4.6. Limitations

Our study has some limitations. We inherited the limitations of all meta-analyses and the limitations of the individual studies of which they were comprised: the varying quality of studies, and multiple variations in study protocols and evaluation [126]. We conducted our meta-analyses only on published articles, so our results were theoretically exposed to a publication bias. We included only studies reporting physicians’ cannabis use, so our results were theoretically exposed to a selection bias. Most cross-sectional studies included in our meta-analysis described a self-report bias. Data were collected by self-administered questionnaire, not always anonymously, which could lead to errors, as it appears in other kinds of self-report questionnaires, where the self-report questionnaire and interview show differences in the answers [127]. Thus, the reporting of cannabis might have been underestimated by physicians. Our meta-analysis also had limitations regarding the definition of cannabis use, although we can assume it is smoking cannabis. The lack of data by specialty precluded further analyses. Comparisons among continents or time periods might also suffer from a different number of studies within each continent or each period. Further studies should assess the prevalence of cannabis use in the general population, which may also permit comparisons with cannabis use in medical doctors.

## 5. Conclusions

Despite a high prevalence (37%) of cannabis use among physicians over a lifetime, daily use is rare but not uncommon, with 1.1‰ medical doctors/students smoking daily. Medical students are the biggest cannabis users. Insufficient data precluded comparisons among medical specialties. Despite the fact that cannabis use is common both in developed and developing countries, there are some cultural differences, with a predominant use in the West. After a decrease in cannabis use since 1990, there has been a rebound since 2005 that should benefit from targeted preventive strategies. Young and male physicians seem to have higher cannabis use, making salient those public health interventions during the early stage of medical studies.

## Figures and Tables

**Figure 1 medicines-10-00029-f001:**
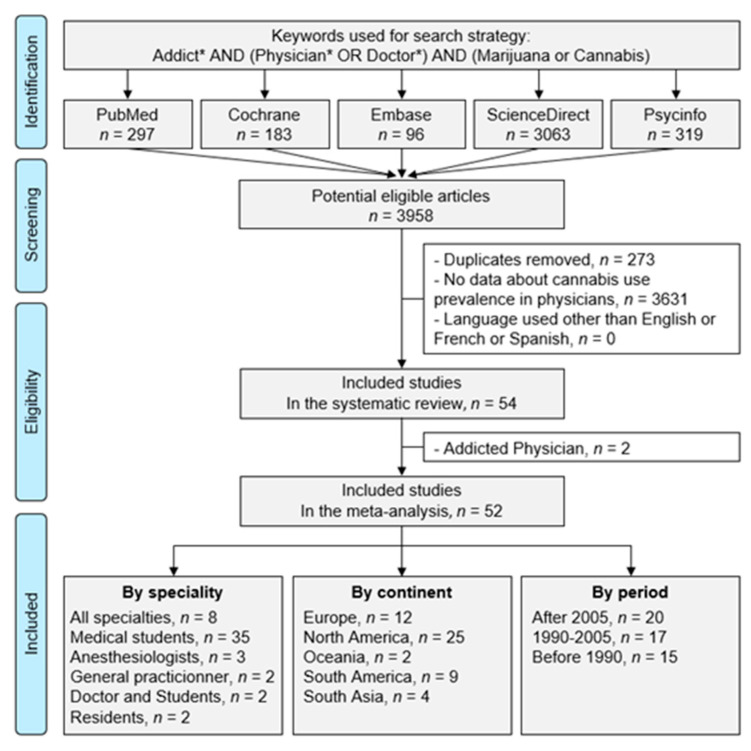
PRISMA Workflow: Search strategy.

**Figure 2 medicines-10-00029-f002:**
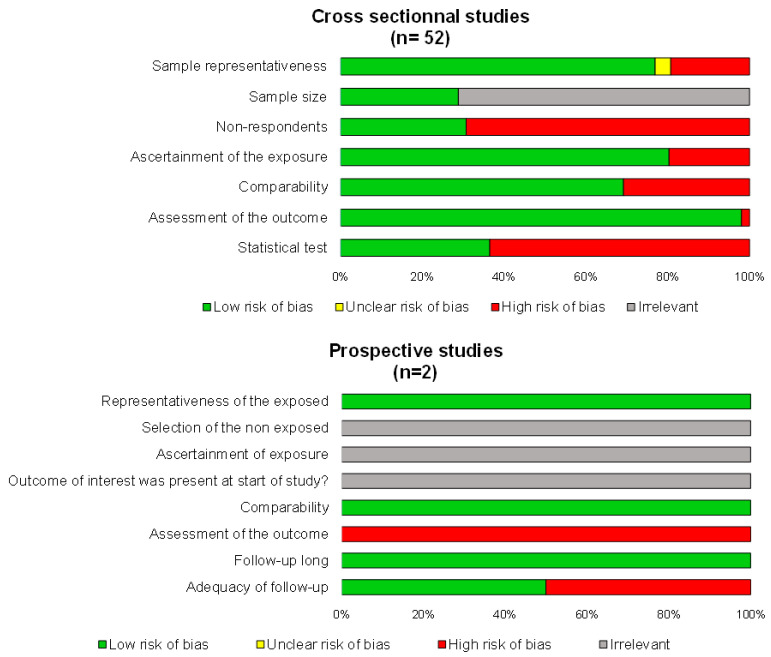
Methodological quality of included articles.

**Figure 3 medicines-10-00029-f003:**
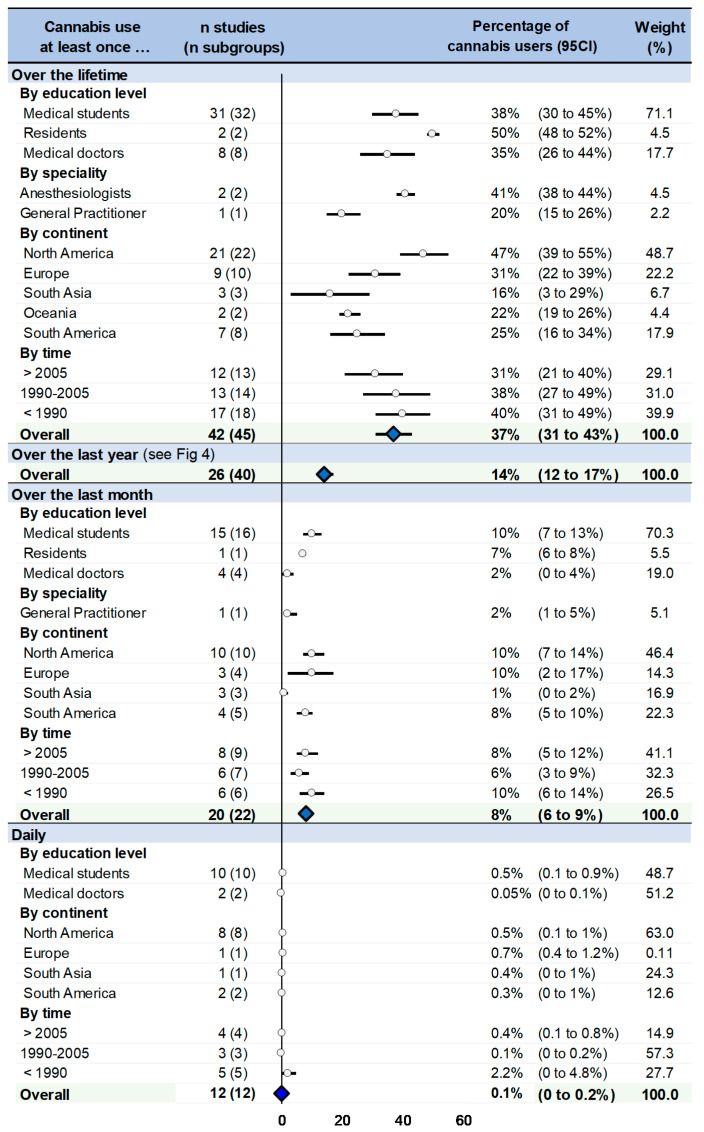
Prevalence of cannabis users among physicians. Each overall summary of a meta-analysis is represented in the graph by a blue lozenge on a horizontal line and each stratification by a dot. Blue lozenges represent the overall percentage of cannabis users. Dots represent percentage of cannabis users by education level, specialty, continent, time. The lozenges / dots represent the overall pooled-effect estimate of individual meta-analyses (pooled effect size—ES), and the length of each horizontal line around the dots represent their 95% confidence interval (95CI). Shorter lines represent a narrower 95CI thus higher precision around pooled-ES. Conversely, longer lines represent a wider 95CI and less precision around pooled-ES. The black solid vertical line represents a percentage of 0% of cannabis users. n studies (subgroups): number of studies and subgroups included for each meta-analysis; I-squared (%): percentage of heterogeneity between studies for each meta-analysis; Weight (%): Weight of each stratification (meta-analysis) in the overall meta-analysis.

**Figure 4 medicines-10-00029-f004:**
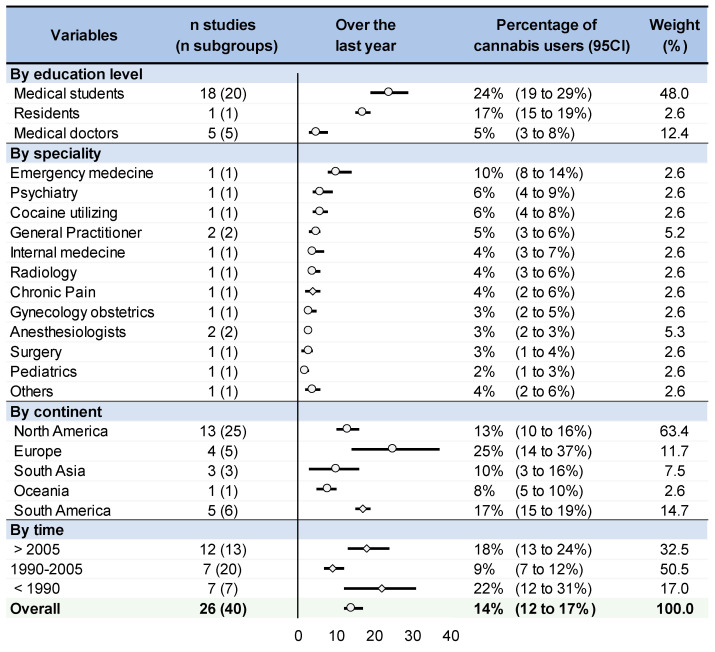
Prevalence of cannabis users among physicians over the past year. Each overall summary of a meta-analysis is represented in the graph by a dot on a horizontal line. Dots represent percentage of cannabis users over the past year by education level, specialty, continent, time, and the length of each horizontal line around the dots represent their 95% confidence interval (95CI). The black solid vertical line represents a percentage of 0% of cannabis users over the past year. n studies (subgroups): number of studies and subgroups included for each meta-analysis; I-squared (%): percentage of heterogeneity between studies for each meta-analysis; Weight (%): Weight of each stratification (meta-analysis) in the overall meta-analysis.

**Figure 5 medicines-10-00029-f005:**
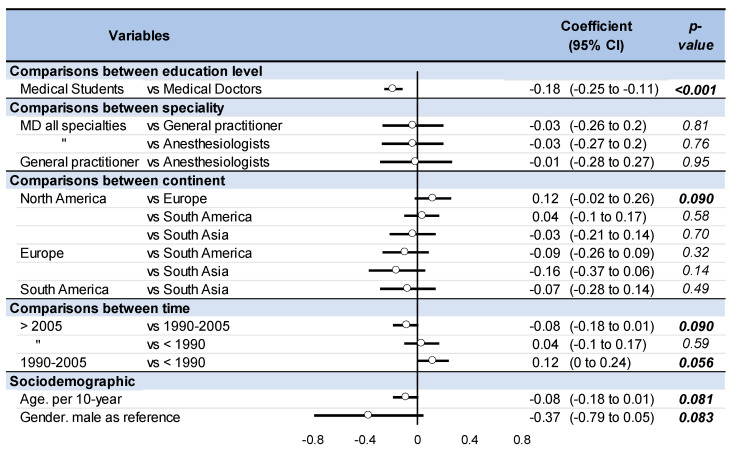
Meta-regressions and factors influencing prevalence of cannabis users among physicians over the past year. The effect of each variable on the outcome (i.e., prevalence of cannabis users among physicians over the past year) is represented in the forest-plot by a dot on a horizontal line. The dots represent the coefficient for each variable, and the length of each line around the dots represent their 95% confidence interval (95CI). The black solid vertical line represents the null estimate (with a value of 0). Horizontal lines that cross the null vertical line represent non-significant variables on the outcome.

**Table 1 medicines-10-00029-t001:** Characteristics of included studies. * Included only in the systematic review.

Study	Country	Data Collection		Physicians		Specialty				Prevalence	(%) of	Cannabis	Users		
			n	% Men	Age	Overall	By Specialty	Residents	Students	Lifetime	Past Year	Past 6 Months	Last Month	Last Week	Daily Use
Ashton 1995 [51]	England	1994	185	41.3	20.4				X	49.2					
Ayala 2017 [36]	USA	2015–2016	855	35.5	25.6				X		26.2		11.7		
Baldwin 1991 [44]	USA	1987	2046	62.7	27.7				X	66.4	22.5		10		
Baptista 1993 [49]	Venezuela	1990	191	49.7	31			X		7.3					
Bazargan 2009 [37]	USA		763	75.1	53	X					4				
Beaujouan 2005 [34]	France	2001	3453	63.3			Anesthesiologists				2.6				
Boniatti 2007 [52]	Brasil	2006	183	45.9	22.5				X	31.1	13.7		7.7		
Carvalho 2008 [53]	Brasil	2005	465	57	21.5				X	14.4					
Chan 2017 [54]	USA	2014	236	52	30				X	53.8					
Coleman 1997 [23]	USA	1989–1991	152						X	32				3	
Conard 1988 [55]	USA	1975–1985	589	65	27.6				X	73.7	31.6		17.3		1.2
Cottler 2013 * [20]	USA	2008–2009	99	76.7	45.6	Addicted	X				29.2				
Croen 1997 [71]	USA	1991–1993	170	54.1					X		29.4				
Da Silveira 2008 [56]	Brasil	2007	456	54.2	21				X		16.4				
De Oliveira 2009 [57]	Brasil	1996–2001	248	52					X	27.3	20.2		13.8		
De Sousa2021 [58]	Brasil	2020	978	65.6			Anesthesiologists			43.2					
Engs 1980 [50]	Australia	1980	431						X	18.3	7.5				
Farrell 2019 [40]	New Zealand	2019	220	29.5					X	35					
Gignon 2015 [59]	France	2011	171	43.8	22.1				X		77		14		7
Hughes 1991 [41]	USA	1987	1785	70	30			X		65.1	17		7		0.3
Hughes 1992 [48]	USA	1990	5426	82.2		X	X			35.6	4.6 to 10.5		2.1 to 7		0.1 to 0.3
Kenna 2004 [35]	USA	2002	104	74	49	X				51.9	3.8		1.9		0
Kory 1984 [8]	USA	1980	165	71.5					X	75.8	43.6		21.8	13.4	8.5
Lambert Passos 2006 [60]	Brasil	1998	1054	47.4	21.1				X	20.9			5.6		
Laporte 1977 [61]	Spain	1974	808	63.7					X	9.6					
Laure 2003 [27]	France		202	75.7	45.6		General practioner			20			2		
Linn 1990 [38]	USA	1987	303	87.1	47.6	X				55		9			
Lipp 1971 [32]	USA	1970	1063						X	49.8	29.9				
Lipp 1972 [47]	USA	1971	1314			X				25					
Lutsky 1993 [39]	Canada		183	84.5		X	Anesthesiologists			30					
Lutsky 1994 [45]	Canada		824	91.9		X				16.2					
Mansky 1999 * [19]	USA		576			Addicted					1.6				
McAuliffe 1984 [21]	USA	1981–1982	134	76.1					X	61	28				
McKay 1973 [62]	Scotland	1971	749	68.7					X	13.3					
Merlo 2017 [30]	USA	2014	862	42.8					X	46.8				4.1	1.5
Newburry-Birch 2001 [70]	UK	1995–1999	122	34.4	58.3	X			X	46.8–65.5	21.9–23.6		11.4–11.8	2.7–7	
Petroianu 2010 [63]	Brasil		332	48.2					X		15.6			0.6	0.3
Pickard 2000 [64]	UK		46	33.8					X	33.5					
Polakoff 1972 [46]	USA	1969	395			X			X	13 to 42					
Rai 2008 [43]	India	2003	2135	70.8	20.5				X	6.6			1.5		
Rochford 1977 [65]	USA		134						X	68.7					
Rodriguez 1986 [42]	Spain	1984	2308	48.1					X	20.7		0.9			
Romero 2009 [26]	Chile	2005	569	55	21.5				X	33	19.7		5.1		0.17
Saeys 2014 [31]	Belgium	2011	626	57.3	45		General practioner				4				
Schwartz 1990 [66]	USA	1987	263	64.6					X	43			5	3.5	1.5
Shyangwa 2007 [22]	Nepal		193	67.3	22.8				X		15				
Singh 1979 [28]	India	1976–1977	672	79.9					X	23.2	11		1.3		0
Singh 1980 [29]	India	1977–1978	95	75.7		X				20	3.2		0		
Slaby 1971 [25]	USA	1970	46	89.1	25.7				X	52					
Solursh 1971 [24]	Canada and USA		234						X	54.3		32.1	22.7	8.5	1.3
Vujcic 2017 [67]	Serbia	2015	418	37.3	22.5				X	34.9					
Webb 1998 [68]	UK	1996	785	44	20				X	43.9				9.6	
Zhou 2015 [69]	USA	2014	431	50.3	25				X	31.1	12.1		8.9		

The cross X is used to show wich kind of specialty the studies are treating, overall/medical specialty/residents/students.

## Data Availability

All relevant data are within the paper.

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
