# Peer review of "Cannabis Use in Physicians: A Systematic Review and Meta-Analysis"

_medicines, 2023, doi:10.3390/medicines10050029_

Round 1
Reviewer 1 Report
This paper describes the use of cannabis among physicians, doctors and medical students in different continents. It is actually good paper- Authors selected only the most important papers, discussed them in details and made relevant conclusions. For me paper is ready for publication.
Author Response
Thank you for your very positive comment
Reviewer 2 Report
This is a good review and analysis which could be improved by adding some other work in the field. I think that , in general, the researchers in impaired MDs or health professionals have shown that physicians may have similar alcohol use than non MD peers but more opioid and other drug use. Also, in the USA it has been shown that gender differences that were significant when women were underrepresented in medicine are disappearing or have disappeared. The work of Dupont , Mclellan and Merlo are worth reviewing in more detail to understand the prevalence of discussions in the literature. Also, more anesthesiologists' findings with SUDs have been reported for many years and are often attributed to secondhand exposures in the OR. Nevertheless, reviewing the anesthesiology vs. other MD types in this regard is worthwhile. In the USA while cannabis might be used medicinally for depression or sleep or similar. This was recently reviewed in the American Journal of Psychiatry and the APA Council by Harvard's Kevin Hill. Lastly, why younger medical students or other MDs use is conjecture, and no data is available to evaluate the competing hypotheses. The idea is that cannabis use stimulates more use or that cannabis users might. Lastly and more significantly, depression or anhedonia might cause more use or use. Blum's work on cannabis and deprssion are important here. Cannabis use by physicians is relevant to fitness for duty concerns similar to those for other drugs and alcohol.
Author Response
Thank you for your comment. The objective of our systematic review and meta-analysis was to assess the prevalence of cannabis use by medical doctors (MDs)/students. We totally agree with you that comparisons with non-MD peers may be of particular interest. We added the following sentence in the limitations: “Further studies should assess the prevalence of cannabis use in the general population, that may also permit comparisons with cannabis use in medical doctors.” Following your suggestion, we read with great interest the work from authors such as Dupont, Mclellan and Merlo. We added some sentences in the discussion such as: “Cannabis can also exacerbate pre-existing psychosis (REF from Merlot ; see PMID 17073176)”; “If general practitioners are heaviest tobacco smokers [10], data are lacking for cannabis. To our knowledge, there are no data on co-addiction in medical doctors, i.e. combination of smoking, alcohol, cannabis or other psychoactive drugs. As alcohol use may predict cannabis use, particularly in youngest (REF from Dupont 34236277), a longitudinal follow-up may be of particular interest”; “No pharmacotherapy treatments demonstrated efficacy – from nicotine replacement therapy to psychotropic drugs [88] –, but there are effective psychosocial interventions [88], such as completing goals self-fixed [86,89]. Very interestingly, targeting the microbiome as a therapeutic and diagnostic tool may also be a promising avenue of exploration in the forthcoming years, considering the role of the gut-brain axis in a wide range of Substance use disorders (REF from Mclellan PMID 34694571)”; “In USA, the States that legalized medical usage of cannabis saw an increase of illicit cannabis use [111]. The consequences of legalisation of cannabis is still under debate. As medical use of cannabinoids become more available, the need for an evidence base evaluation of safety and efficacy is necessary (REF from Kevin P Hill PMID 34875873)”; “Medical students may also use cannabis in university to cope with stress or depressive episodes [94]. However, cannabis decreases memory function of students (and cannabis abstinence leads to improved memory), and is associated with poorest academic performance [94], that could be related to the cannabis-induced hypodopaminergic anhedonia (REF from Blum PMID 33868044)”; “The moment of use may be relevant considering putative side effects of cannabis on medical errors. Cannabis use by physicians is relevant to fitness for duty concerns similar to those for other drugs and alcohol (PMID 29280776; PMID 8366391; PMID 23860546).”